# No Factor Left Behind:
# Towards arbitrary amount of factors in the medical cohort analysis

## Abstract

Driven by the goal of data-driven analysis on the large-scale cohort, a large language model(LLM) has solidified itself as a critical focus of artificial intelligence medical research today. However, such efforts have coalesced around a small group of evidence, leaving behind the vast majority of factors collected in the cohort investigation. What does it take to break the more than 70 factors while ensuring responsible, high-quality prediction, all while keeping medical considerations in mind? In No Factor Left Behind, we first took on this challenge by numerical interpretable evidence contextualizing the need for Premature rupture of membranes (PROM) risk assessment through exploratory interviews with domain experts. Then, we created datasets and models aimed at narrowing the performance gap between low and high-frequency factors. More specifically, we developed a model based on factor-value pairs trained on data obtained with robust and effective data mining techniques tailored for low-frequency factors. We propose multiple architectural and training improvements to counteract overfitting while training on 70 factors. Critically, we interpreted the risk of PROM over 7000 cohort participants' directions using numerical interpretable evidence with precise values of factors combined with human evaluation covering all factors in the dataset to assess medical safety. Our model achieves a performance of 79% accuracy (78 factors) and 96% accuracy(40 factors) with risk assessment at the screening level, laying the novel insight for realizing a general medical cohort analysis method in the era of LLMs.

## 1 Introduction

There may not exist another domain like medical cohort analysis that requires both a high level of expert knowledge and substantial human resources while acquiring expert-interpreted data is quite expensive. Medical cohort studies involve the systematic collection and analysis of vast amounts of heterogeneous data, encompassing clinical measurements, demographic information, genetic data, lifestyle factors, and more. The integration and interpretation of these diverse factors are crucial for understanding disease mechanisms, predicting patient outcomes, and personalizing treatment strategies. Traditionally, medical cohort analyses have focused on a limited set of well-established factors, often driven by prior clinical knowledge or the availability of high-frequency data. While this approach has yielded significant insights, it inherently overlooks a multitude of potentially relevant factors that may have low prevalence or are less studied. Ignoring these factors can lead to incomplete models that fail to capture the complexity of medical conditions, potentially missing critical predictors of patient outcomes.

Recent advancements in artificial intelligence, particularly the development of large language models (LLMs), have opened new avenues for data-driven analysis in healthcare. LLMs excel at handling large-scale, high-dimensional data and can uncover complex patterns that traditional statistical methods might miss. Currently, medical LLMs have made significant strides in enhancing clinical decision support, medical documentation, and patient interaction. Models like Biollama and ClinicalBERT have demonstrated improved performance in tasks such as disease classification, symptom extraction, and electronic health record (EHR) analysis (Kraljevic et al., 2021; Saab et al., 2024; Wu et al., 2023). Additionally, specialized LLMs are increasingly being integrated into diagnostic tools,

enabling more accurate and timely predictions (Qin et al., 2023). These advancements underscore the potential of LLMs to transform healthcare by providing deeper insights and supporting more informed medical decisions. However, the application of LLMs in medical cohort analysis has predominantly concentrated on a narrow set of evidence, leaving the vast majority of collected factors underutilized. This imbalance not only limits the predictive power of models but also restricts the discovery of novel insights that could emerge from a more comprehensive analysis.

In this paper, we aim to leverage the powerful pre-trained large language models like llama3.1 series and Phi3.5 MoE with expressive medical prompts to make efficient domain transfers from natural language to medical language for risk assessment. To this end, we first explore how to manually design effective medical prompts by using hierarchical prompt with Chain of thought(CoT), and show that such well-designed prompts can significantly improve the domain transfer risk assessment compared to the default factors names and values. Intuitively, the common factors' names in text prompts, such as education level, sleeping time, and clinical measurements, are different aspects of participants, and therefore, by clustering factors to these expressive attributes in the prompts, the LLMs can selectively learn to align features' meaning with value in the prompts rather than aimlessly learning.

Furthermore, to improve the efficiency and avoid the laborious manual annotations, we propose several approaches, i.e., masked language model (MLM) auto-prompt generation with numerical feature interaction map, factors' knowledge specific auto-prompt generation or a hybrid of both, to automatically generate medical prompts that make the LLMs perform on par with the model with manually elaborated prompts. The MLM-driven approach mainly focuses on extracting expert-level knowledge from pre-trained language models specialized in the medical cohort domain. In contrast, the cohort-specific prompt generation, based on the Table question answering (TableQA) system, allows the flexibility in designing prompts to include cohort-specific attribute information rather than using a single fixed prompt for all participants during inference.

We evaluate our approaches on a wide range of existing open-source models across different arch, context window, and parameter sizes. The models with our well-designed medical cohort prompts exhibit significant superiority over those with default prompts in terms of zero-shot and few-shot performance, some surpassing the supervised model trained with full data. Moreover, our fine-tuned models outperform the traditional supervised baselines by a significant margin across almost all models.

## 2  RELATED WORK

In this section, we review the existing literature pertinent to our work, focusing on five key areas: transfer learning between natural and medical language domains, prompt design in language models, table question answering (TableQA), retrieval-augmented generation (RAG), and the integration of external tools through techniques like Toolformer.

### 2.1  TRANSFER LEARNING BETWEEN NATURAL AND MEDICAL LANGUAGE DOMAINS

Transfer learning has become a prevalent strategy for training deep neural networks in domains with out-of-distribution data, such as the medical field. In natural language processing (NLP), models pre-trained on large-scale general-domain corpora are fine-tuned on domain-specific datasets to adapt to specialized vocabulary and concepts. This approach is particularly valuable in the medical domain, where annotated data is scarce and expensive to obtain due to the need for expert interpretation. Several studies have explored the transfer of linguistic knowledge from natural to medical language domains. For instance, BioBERT (Lee et al., 2019) and ClinicalBERT (Alsentzer et al., 2019) are adaptations of BERT (Devlin et al., 2019), pre-trained on biomedical and clinical text corpora, respectively (Singhal et al., 2022). These models have shown significant improvements in various medical NLP tasks, including named entity recognition, relation extraction, and question answering(Zhao et al., 2023; Yang et al., 2022c;b). However, most of these models focus on high-frequency medical terms and conditions, potentially overlooking low-frequency but clinically significant factors. Our work addresses this gap by developing models capable of integrating a broader range of factors from medical cohorts.

## 2.2 PROMPT DESIGN

Knowledge-intensive domains like medicine require language models to comprehend and generate domain-specific content accurately. Prompting techniques have emerged as a way to guide language models in generating desired outputs by framing tasks as text-completion problems. Effective, prompt design is crucial for eliciting the correct information from language models, especially when dealing with specialized knowledge. Recent advancements have introduced methods like prompt tuning (Lester et al., 2021) and instruction-based learning (Mishra et al., 2022), which fine-tune language models with minimal additional parameters or adapt them using natural language instructions. In the medical domain, prompt design helps models interpret complex clinical queries and generate responses that are both accurate and contextually appropriate (Wang et al., 2024; Zaghir et al., 2024). Our approach leverages prompt design to enhance the interpretability and reliability of risk assessments, ensuring that all factors in the cohort are considered.

## 2.3 TABLE QUESTION ANSWERING

TableQA involves interpreting structured data and answering queries based on the information contained within tables. Large language models have shown promise in comprehending and analyzing tabular data, which is crucial for medical cohort analysis, where patient data is often stored in tabular form. Models like TaBERT (Yin et al., 2020) and TAPAS (Herzig et al., 2020) have been developed to jointly encode tables and text, enabling them to perform tasks like table-based question answering and fact verification (Zhang et al., 2024; Zha et al., 2023) . These models integrate the structural information of tables with textual data, allowing for more nuanced understanding. However, privacy concerns in medical data limit the use of proprietary models. Our work builds upon open-source LLMs to process cohort data effectively meeting the biosafety and privacy constraints while providing a practical method for medical cohort data analysis.

## 2.4 RETRIEVAL-AUGMENTED GENERATION (RAG)

RAG is a methodology that enhances language models by providing them with direct access to external knowledge bases during the generation process. By retrieving relevant information, models can produce outputs that are more accurate and informative, especially in domains where up-to-date or specialized knowledge is essential (Li et al., 2024). In the context of medical cohort analysis, RAG can help reduce hallucinations—instances where the model generates incorrect or nonsensical information and enhance reasoning abilities in risk assessments. Lewis et al. demonstrated that incorporating retrieval mechanisms allows models to generate more factual responses (Lewis et al., 2021). Our study builds upon RAG by embedding information from the cohort population and participant factors, thereby improving the model's ability to consider all relevant factors and produce more reliable risk assessments.

## 2.5 TOOLFORMER

The Toolformer technique enables large language models to leverage external tools through self-supervised learning (Schick et al., 2023). By training the model to determine which APIs to call, when to call them, and how to integrate the results, Toolformer extends the capabilities of LLMs beyond text generation. Our study utilizes these advancements by training an LLM to incorporate machine learning-based information into natural language risk assessments (Lundberg & Lee, 2017). This approach enhances both the robustness and interpretability of the screening process without the need for expert annotation, thereby streamlining the analysis and making it more scalable. By integrating external computational tools, the model can perform complex calculations and access up-to-date data, which is critical for accurate medical assessments.

## 3 METHODOLOGY

In this work, we mainly explore how to leverage the entailed cohort knowledge and experience in the large language models, such as llama3 and TableLLM (Dubey et al., 2024; Zhang et al., 2024), and transfer it to medical domains. Towards this end, we conduct a comprehensive study on a variety of risk assessment tasks in medical cohort domains, where we propose several strategies for

better elicitation of medical knowledge from large language models pre-trained on natural language. We focus on the design and automatic generation of medical prompts that can include expert-level knowledge and cohort-specific information, which empowers the large language models for health risk assessment in both zero-shot transfer and fine-tuning conditions.

## 3.1 PRELIMINARIES

Unifying tabular data and language pre-training norms have emerged as a powerful approach to improve LLM performance in various table-related tasks, showcasing promising cross-domain transfer capabilities. Inspired by the success of incorporating language supervision in visual recognition, TableLLM adopts a similar philosophy by integrating textual prompts with tabular data. For instance, when dealing with spreadsheet-embedded tabular data, TableLLM receives both the table header and a subset of rows alongside a text prompt specifying the desired manipulation operation. This prompt can take the form: Prompt = "[Operation]-[Subcategory] Instruction", where [Operation] denotes the main operation type (e.g., Query, Update, Merge, Chart) and [Subcategory] specifies the sub-operation (e.g., Filter, Aggregate, Sort). This integration allows TableLLM to leverage the rich semantic information embedded within natural language instructions to effectively understand and execute complex tabular data manipulations. It is not hard to see that the data-text inputs have been sufficiently aligned, so one could provide an auxiliary prompt input to guide the LLMs to reasoning the factors' value and association more easily. Given that, we believe a well-designed prompt could largely enhance the performance of the pre-trained models on the table-related tasks, especially in an unfamiliar domain like the medical cohort

## 3.2 MEDICAL PROMPT DESIGN WITH HIERARCHICAL PROMPT WITH CHAIN OF THOUGHT(CoT)

Here, we take the TableLLM model as an entry point to explore how to utilize the text prompts and large language models entailed knowledge to bridge the gap between the natural and medical language domains smoothly. Similar to previous findings in natural language (Yang et al., 2022a; Iida et al., 2021), our preliminary experiments also indicate that providing an expressive description in medical prompt can primarily benefit the zero-shot transfer performance of large language models in out of distribution medical data. More importantly, we find that the annotation of cohort factors in medical domains could significantly increase the amount of the factor during the risk assessment to become more comprehensive and robust.

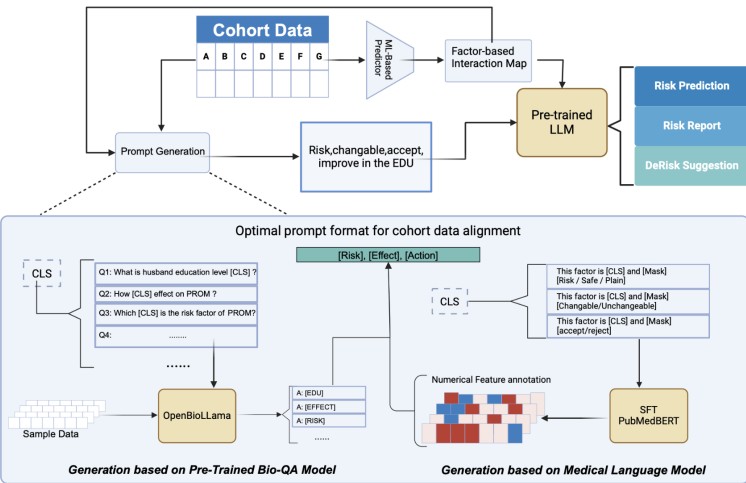

Figure 1: Overview of the proposed approach. The optimal medical prompts can be automatically generated with the help of a pre-trained OpenBioLLM model, a medical language model, or a hybrid of both.

$$\text{Prompt} = \sum_m \text{Template} \left[ (V_i, \text{factor}_m), \text{Label}(\text{Factor}_i, \text{Value}_i), \text{Interaction}(\text{Factor}_i) \right], \quad (1)$$

where: $\text{Factor}_i$ represents the individual factors influencing the prompt. Template are the predefined text structures that incorporate these factors. $V_i$ are variables or specific values associated with each factor. $\text{Label}(\text{Factor}_i, \text{Value}_i)$ denotes the labeling of each factor and its value for better traceability in the prompt generation process. $\text{Interaction}(\text{Factor}_i)$ captures the potential interactions between different factors, which could affect the final output of the prompt.

Following this idea, we propose to design medical prompts with a focus on the hierarchy and interaction of factors describing the medical cohort of interest. Assuming M amount of cohort factors where the summation means the concatenation of M factors of cohort annotates by three steps. For example, the factor-value pair of husband education will be extended by expert-level knowledge from systemic review and tutorial to detailed contextualize in general effect, risk and meaning. Moreover, the value of the husband's education will be claimed as [risk, unchangeable, accpet]. By annotating the specifically engineered attributes, the zero-shot results increase significantly and surpass the results of providing only the default factors' names by a large margin. This pattern could be seen in a variety of large language models across parameter size and architecture from llama3.1 to Phi3.5 MoE, demonstrating the effectiveness of well-designed medical prompts with hierarchical prompts with Chain of thought.

However, during the process of searching for appropriate prompts, we also find that the current text prompt design has the following limitations: Firstly, manually designing an effective prompt requires expert-level knowledge and personal bias on the human experts are difficult to control; Secondly, in the current large language models, the prompts are normally fixed for all samples during inference,, which is not ideal for large scale cohorts that have varying participants. For example, the pregnant participants often have diverse domestic backgrounds and behavior patterns

### 3.3 FACTOR-BASED INTERACTION MAP

The contextual information from the knowledge provided expert insight. However, following the epidemiological reasoning theory, we propose the factor-based interaction map based on the game theory focusing on providing information on the relationship among the factors as equation 3 shows and the effect on the PROM(2).

$$\phi_i(v) = \sum_{S \subseteq N \setminus \{i\}} \frac{|S|!(|N| - |S| - 1)!}{|N|!} [v(S \cup \{i\}) - v(S)]. \quad (2)$$

where: $\phi_i(v)$ represents the contribution of factor $i$ to the overall value, assessed through the value function $v$. $N$ is the set of all factors considered in the model. $S$ is any subset of $N$ that excludes factor $i$. $|S|$ denotes the cardinality (number of elements) of subset $S$. $|N|$ is the total number of factors in the set $N$. $v(S)$ is the value function representing the outcome when only the factors in subset $S$ are present. The term $v(S \cup \{i\}) - v(S)$ captures the marginal contribution of adding factor $i$ to subset $S$.

$$I_{ij} = \frac{1}{2} \sum_{S \subseteq \{1, \dots, p\} \setminus \{i,j\}} \frac{|S|!(p - |S| - 2)!}{p!} [v(S \cup \{i, j\}) - v(S \cup \{i\}) - v(S \cup \{j\}) + v(S)],$$

$$(3)$$

where: $I_{ij}$ represents the interaction value between factors $i$ and $j$. $p$ is the total number of factors considered in the model. $S$ is a subset of factors, specifically excluding factors $i$ and $j$. $|S|$ denotes the cardinality of subset $S$, i.e., the number of elements in $S$. $v(S)$ is the value function representing the predicted outcome when only the factors in subset $S$ are considered. The term within brackets quantifies the incremental prediction change when both factors $i$ and $j$ are considered together compared to when they are considered separately.

## 3.4 AUTOMATIC GENERATION OF MEDICAL PROMPTS

To overcome such limitations, in this section, we further investigate how to efficiently generate knowledge-rich and value-specific prompts. Particularly, we discuss about the creative auto-prompt pipelines we proposed for generating expert-level knowledge supported and table-specific prompts.

Masked Language Model Driven Auto-Prompt Generation

Masked Language Model Driven Auto-Prompt Generation To obtain expert-level knowledge, we utilize medical knowledge in expert-level BERT-like pre-trained language models, e.g., the Pub-MedBERT model (Gu et al., 2021; Devlin et al., 2019) , for annotating factor-value pair of a medical concept. Since the model's weight was released in 2021, We've used supervised fine-tuning on Pub-MedBERT with an updated dataset that includes geo-specific PROM tutorials, systematic reviews, and original research from reputable journals. This approach aims to tailor the model to the infant health cohort study.

Figure 1 (right side) illustrates the overall flow of our MLM-driven auto-prompt generation pipeline. We first ask the model, which contains medical domain-specific knowledge, to predict the masked token in the given cloze sentences we design. The template of the cloze sentences is given as: 'The [Value] of an [Factor] is [MASK],' where the 'Value' and 'Factor' tokens are provided and represent the factor name and value, respectively. This operation could be formulated as:

$$v^{\text{Val}} = \arg \max_{\tilde{v}^{\text{Val}} \in V} P_{\text{Expert}}([\text{mask}] = \tilde{v}^{\text{Val}} | t_s), \tag{4}$$

where: $v^{\text{Val}}$ represents the predicted value for the masked attribute. $V$ is the set of all possible expert knowledge-augmented phrases that can be applied to fill the mask. $t_s$ denotes the tokens that constitute the cloze sentence template, providing the contextual backbone for the prediction. $P_{\text{Expert}}$ is the conditional probability function that estimates the likelihood of each possible augmented phrase being the correct fill for the masked attribute, based on the provided expert knowledge.

We take M rounds by repeating the above process for each factor using the template defined in Eq4, then add the feature interaction map for each participant. The whole process can be formulated as follows:

$$v^{\text{Val}} = \arg \max_{\tilde{v}^{\text{Val}} \in V} P_{\text{Expert}}([\text{mask}] = \tilde{v}^{\text{Val}} | t_s), \tag{5}$$

where: $v^{\text{Val}}$ is the predicted value for the masked attribute. $V$ represents the set of all possible phrases augmented with expert knowledge, from which the prediction is made. $t_s$ are the tokens constituting the cloze sentence template, providing the necessary context for the prediction. $P_{\text{Expert}}$ denotes the conditional probability of the masked attribute value given the expert-augmented phrase in the context of $t_s$.

## 3.5 CONTEXTUAL AUTO-PROMPT GENERATION

Although with the above MLM-driven prompt generation approach, we can successfully generate auto-prompts that are supported by expert-level knowledge, the prompts are still not flexible enough to include cohort-specific information since the cohort data are difficult to become the pre-train data. Therefore, in this section, we further propose a Contextual-specific auto-prompt generation approach by adopting pre-trained table question answering (TableQA) models, e.g., the OpenBioL-Lama model. As demonstrated in Figure 1 (left side), we ask the QA models multiple questions related to the factor iterative. For example, we can ask the model: "What is the husband's education level?". We expect to receive a proper answer from the QA model and take that answer as contextual information. Unlike the MLM-driven approach, we won't ask for an annotated factor-value pair due to the computation time constraint. This process has to be applied to each factor name input to generate factor-specific prompts, which means the corresponding prompt for each factor is aimed at final LLMs to understand the factor may not contained in its previous pre-train data and be well defined. Given a factor input x, the corresponding prompt could be formulated as follows:

$$\text{Prompt} = \sum_m [\text{Template}\{\text{MLM}(\text{Factor}_i, \text{Value}_i)\}, \text{Interaction\_score}(\text{Factor}_i)], \tag{6}$$

where: Prompt is the final text output constructed dynamically based on input factors. $\text{Concat}_i$ represents the concatenation operation over index $i$, which iterates through each factor involved in the prompt generation. $\text{Template}\{\text{MLM}(\text{Factor}_i, \text{Value}_i)\}$ is a template filled by a masked language model (MLM), where $\text{Factor}_i$ and $\text{Value}_i$ are inputs to the model to generate contextually relevant text snippets. $\text{Interaction\_Score}(\text{Factor}_i)$ quantifies the impact or relevance of the factor $i$ in the context of the interaction among multiple factors, enhancing the contextual alignment of the generated text. $\text{Factor}_i$ represents individual elements from the set of all factors $\{\text{Factor}_1, \text{Factor}_2, ..., \text{Factor}_M\}$.

We believe that the domain transfer performance would be improved if we annotate both expert-level knowledge and cohort-specific information in the prompts. However, our preliminary results obtained from the TableQA prompts suggest that certain factors (e.g., lie time ) may not be appropriately answered by the pre-trained LLM. We speculate that the hallucinations given by the LLM can be explained by the fact that most of the medical languages are taken in a quite different environment compared to the natural language, and therefore expecting the LLM pre-trained on natural language in the general purpose to recognize certain factor name or which association of the is in the cohort could be challenging. In this regard, we choose to combine the two above approaches, namely the MLM-driven approach and the Bio-QA based approach for different factors. For example, we can use the Bio-QA models to provide the detailed contextual information of factor names, while for the risk attribute, we obtain it from the masked language model approach. The intuition behind such a combination is that we think the cohort data are low-frequency data during the pre-train process to provide precise information in the prompt, which will be more effective in helping LLM reasoning and staying up to date rather than post-training. We named the prompts generated by this hybrid approach the 'hybrid prompts', while the ones generated by purely Bio-QA based models are the 'Bio-QA prompts'. In this case, the prompt template in Eq5 for 'hybrid prompts' can be updated to:

$$\text{Prompt}_x = \sum_m [\text{Bio-QA}(x), \text{MLM}(x, \text{Attr\_set})], \tag{7}$$

where: Concat denotes the operation of concatenating two text strings, aiming to merge informative outputs into a single prompt. $\text{Bio-QA}(x)$ represents the output from a Bio-QA model (e.g., OpenBioLlama), which provides detailed contextual information about a biological factor $x$. $\text{MLM}(x, \text{Attribute\_set})$ is the output of a Masked Language Model that generates labels or descriptors for the factor $x$ based on a predefined set of attributes such as risk, changeability, and acceptance.

## 4 EXPERIMENTS

### 4.1 SETUP

Model: For a comprehensive study, we collect 10 public models of various types, including parameter size, architecture, and fine-tune state.

Table 1: Comparison of model characteristics

| Model Name | Parameter Size | Architecture | Context Window Size | Fine-Tune |
|---|---|---|---|---|
| LLama3.1 | 8B/70B/405B | Dense | 16K | No |
| MedAlpaca | 7B | Dense | 2K | No |
| PMC-Llama | 7B | Dense | 2K | No |
| Meditron | 7B | Dense | 2K | No |
| Biomistral | 7B | Dense | 4K | No |
| Phi 3.5 | 42B | MoE | 128K | No |
| OpenBioLLM | 8B/70B | Dense | 16k | Yes |

## 4.2 DATASET AND ETHICS CONSIDERATION

The study utilized data from a maternal and infant health cohort in a major city in eastern China. Participants were recruited from three leading medical centers in the region. Inclusion criteria encompassed women aged 18-40, local residents, without communication barriers, and not undergoing assisted reproductive technology. The tabular dataset was structured into four categories: maternal basic information, family background, pre-pregnancy health status, and second-trimester health status. The final cohort comprised 7,199 subjects with 78 features, including 1,483 cases of premature rupture of membranes. For the Numerical Interpretable Evidence data, the feature-outcome relationships were determined using an ensemble model approach. The specific details of the machine learning interpretability pipeline used to derive these datasets will be elaborated in the methods section.

All participants completed a structured interview based on a face-to-face questionnaire that included information on socio-economic and demographic characteristics, health status and lifestyle during pregnancy. Written informed consent was obtained from all pregnant women and the study was approved by the Research Ethics Committee of the authors' research institution

## 4.3 IMPLEMENTATION DETAILS

For our experiments, we use the llama3.1 8B (Dubey et al., 2024) as our base pre-trained model and follow their hyper-parameter choices when transferring to medical language. We train our Pub-MedBERT models using Adam optimizer with base learning rate of $1 \times 10\text{-}7$ for the PubMedBERT, and the weight decay is set to 0.03. We freeze the bottom two layers of the encoder and decay the learning rate by 0.1 when the validation performance plateaus. For the MLM automatic prompt generation, we use the PubMedBERT-large-uncased variant (Tinn et al., 2021) to superviesd fine tune and fill the cloze sentences. Moreover, we use the OpenBioLLM-8B variant to generate the contextual factor information automatically. For the comparison experiments, we use the previous models MedAlpaca (Han et al., 2023), PMC-Llama-7B (Wu et al., 2023), Meditron-7B (Chen et al., 2023),Med42-70b , Biomistral-7B (Labrak et al., 2024), Phi3.5 MoE (Abdin et al., 2024) and llama3.1 405B.

## 4.4 TRANSFER TO ESTABLISHED MEDICAL COHORT

This section demonstrates that the llama3.1 8B model, with the aid of well-designed language prompts, can directly or indirectly transfer to the medical domain with competitive performance. For convenience, we split the cohort datasets into two major categories: risk prediction and risk report. In the following we first give an overview of our fine-tuned models surpassing the supervised baseline. Then, we illustrate the results of the proposed approach on cohort dataset analysis, focusing on the zero-shot scenario. Finally, we discuss the fine-tuning results on the cohort datasets.

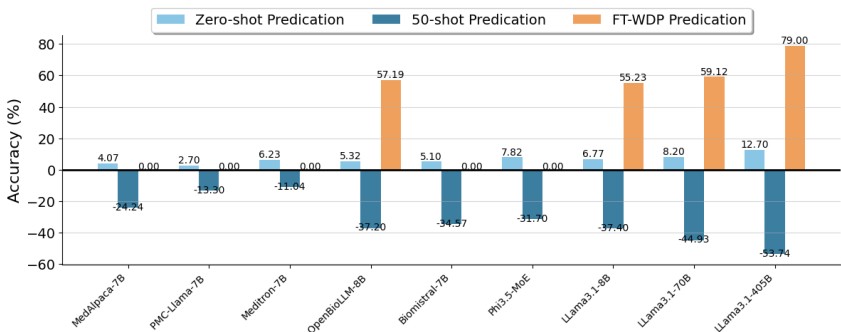

Figure 2: Comparisons with the previous open-source model in 78 factors.

### 4.4.1 TRANSFER PERFORMANCE SURPASSING SUPERVISED METHODS

To prove that text prompts are effective for cohort-domain transfer, we conduct extensive experiments under both zero-shot domain transfer and supervised transfer (post-training) settings. We include a series of supervised baselines: Meditron-7B, Biomistral-7B PMC-Llama-7B and Phi 3.5 MoE for comparisons. As illustrated in Figure 2, our full data fine-tuned models(LLama3.1 405B is not fine-tuned) with well-designed medical prompts (orange) surpass the supervised baseline by a large margin across all models in zero-shot (sky blue) . Moreover, even 50-shot (blue) results are fully surpassed by our method. Interestingly, the well-designed prompt takes the parameter size gap in this task, e.g. OpenBioLLM-8B and LLama3.1-70B.

Table 2: Our approaches v.s. supervised models as the factor's amount increases (Accuracy%)

|  | Model | 20-factors | 40-factors | 78-factors |
|---|---|---|---|---|
| **Zero-shot** | OpenBioLLM-8B | 44.68 | 47.33 | 5.32 |
|  | LLama3.1-8B | 73.17 | 71.08 | 6.77 |
|  | LLama3.1-70B | 79.44 | 76.23 | 8.20 |
|  | LLama3.1-405B | 73.02 | **78.03** | 12.70 |
|  | MedAlpaca-7B | 26.24 | 23.1 | 4.07 |
|  | PMC-Llama-7B | 17.30 | 18.9 | 2.70 |
|  | Meditron-7B | 9.35 | 6.49 | 6.23 |
|  | Biomistral-7B | 36.79 | 31.79 | 5.10 |
|  | Phi3.5-MoE | **80.85** | 74.35 | 7.82 |
| **50-shots** | OpenBioLLM-8B | 55.82 | 57.71 | 37.20 |
|  | LLama3.1-8B | 79.20 | 80.19 | 37.40 |
|  | LLama3.1-70B | 86.70 | 81.80 | 44.93 |
|  | LLama3.1-405B | 86.42 | **85.12** | 53.74 |
|  | MedAlpaca-7B | 30.60 | 34.68 | 24.24 |
|  | PMC-Llama-7B | 19.74 | 19.7 | 13.30 |
|  | Biomistral-7B | 38.60 | 37.97 | 34.57 |
|  | Phi3.5-MoE | **81.02** | 79.60 | 31.70 |
| **Prompt-Assigned Zero-shot** | OpenBioLLM-8B (Hybrid) | 92.02 | **96.12** | 57.19 |
|  | OpenBioLLM-8B (Manual) | 89.00 | 85.00 | 65.00 |
|  | LLama3.1-8B (Hybrid) | **93.70** | 94.31 | 55.23 |
|  | LLama3.1-70B (Hybrid) | 92.52 | 93.90 | 59.12 |
|  | LLama3.1-405B (Hybrid) | 92.20 | 95.17 | **79.00** |

Table 2 shows the quantitative numbers for each factor's increase. Figure 4 also supports this, showing that the LLMs significantly outperform the classical risk assessment with fully supervised learning, especially in high-dimensional settings.

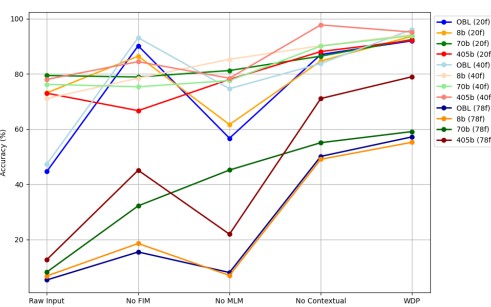

Figure 3: Contextual annotation with feature interaction in the prompts improves the risk prediction as the factors amount increased.

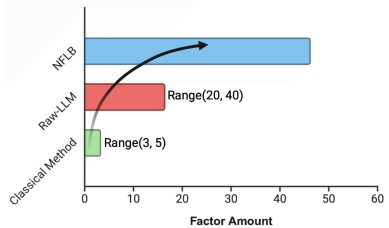

Figure 4: Data efficiency comparison between No Factor Left Behind(our method) and classical risk assessment models (logistical regression.)

The effectiveness of annotation and auto-prompts In section 3.2, we discussed that adding annotation could make the models perform better in zero-shot tasks. Here, we demonstrate in Figure 3 an overall pattern of the effect of attribute injection on performance under the zero-shot setting. As shown in the figure, the overall performance increases as more information is integrated into the prompts. This is also illustrated in Figure 3, where various annotation and their combinations are shown to improve the results. As this process is rather tedious and time-consuming, we need qualified automatic approaches to accelerate the generation process and scale it up without sacrificing too much performance. Fortunately, the models with our proposed auto-prompts, especially with the hybrid and MLM-driven approaches, show comparable results to those with manually created prompts and surpass those with default prompts by a landslide. Figure 5 shows an example of the auto-prompt generation with the hybrid approach of 40 factors.

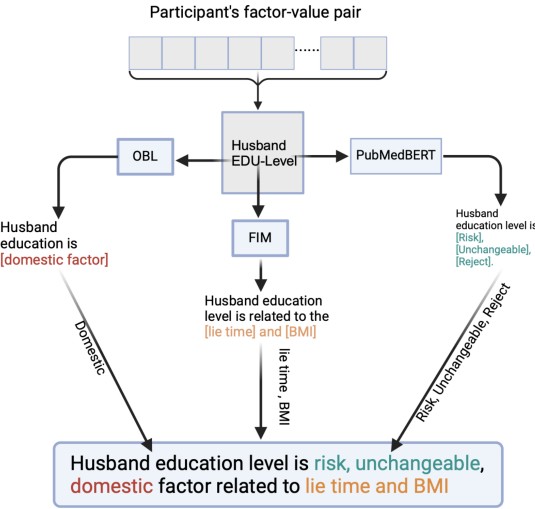

Figure 5: Auto-prompt generation showcase

## 4.5 ABLATION STUDIES

Table S16 12presents the ablation studies on the prompt engineering of hierarchical prompts with CoT for 40-factor and 78-factor situations. As shown in the table, our default choice of using hierarchical prompts with CoT has a higher rate of detecting PROM cases. The hierarchical prompt provides a better understanding of the 78 factors, and we find that the CoT has little impact on overall accuracy but greatly helps to identify the normal case.

## 5 CONCLUSION

This paper comprehensively studies how to leverage the large-scale large-language models pretrained on general language tasks to the medical cohorts. We present that well-designed medical prompts containing domain-specific knowledge are the key to bridging the gap between domains. Therefore, we propose several approaches to generate medical prompts manually or automatically. While the manual approach tremendously improves the zero-shot performance compared to the default prompts with object names, the automatic approaches allow us to generate expert knowledge augmented and cohort-specific prompts on a large scale. Extensive experiments are conducted on the 11 different medical models across various aspects, showing that the prompts generated by our approaches can improve the transfer performance, and our fine-tuned models surpass the supervised baselines by a large margin. This superior domain transfer performance also prompts us to explore more cohort-efficient language algorithms to benefit medical cohort understanding.

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

## A  APPENDIX

| | Model | Accuracy | TPR | TNR | Precision | Recall | F1 |
|---|---|---|---|---|---|---|---|
| **Zero-shot** | OpenBioLLM-8B | 44.68 | 0.69 | 0.38 | 0.23 | 0.69 | 0.34 |
| | LLama3.1-8B | 73.17 | 0.6 | 0.77 | 0.4 | 0.6 | 0.48 |
| | LLama3.1-70B | 79.44 | 0.84 | 0.78 | 0.5 | 0.84 | 0.63 |
| | LLama3.1-405B | 73.02 | 0.42 | 0.81 | 0.36 | 0.42 | 0.39 |
| | MedAlpaca-7B | 26.24 | 0.52 | 0.19 | 0.14 | 0.52 | 0.23 |
| | PMC-Llama-7B | 17.3 | 0.15 | 0.18 | 0.04 | 0.15 | 0.07 |
| | Meditron-7B | 9.35 | 0.19 | 0.07 | 0.05 | 0.19 | 0.08 |
| | Biomistral-7B | 36.79 | 0.59 | 0.31 | 0.18 | 0.59 | 0.28 |
| | Phi3.5-MoE | 80.85 | 0.67 | 0.84 | 0.53 | 0.67 | 0.59 |
| **50-shots** | OpenBioLLM-8B | 55.82 | 0.61 | 0.55 | 0.26 | 0.61 | 0.36 |
| | LLama3.1-8B | 79.2 | 0.21 | 0.94 | 0.49 | 0.21 | 0.29 |
| | LLama3.1-70B | 86.7 | 0.76 | 0.89 | 0.65 | 0.76 | 0.7 |
| | LLama3.1-405B | 86.42 | 0.5 | 0.96 | 0.76 | 0.5 | 0.6 |
| | MedAlpaca-7B | 30.6 | 0.64 | 0.22 | 0.17 | 0.64 | 0.27 |
| | PMC-Llama-7B | 19.74 | 0.51 | 0.12 | 0.13 | 0.51 | 0.21 |
| | Biomistral-7B | 38.6 | 0.71 | 0.3 | 0.21 | 0.71 | 0.32 |
| | Phi3.5-MoE | 81.02 | 0.69 | 0.84 | 0.53 | 0.69 | 0.6 |
| **Prompt-Assigned Zero-shot** | OpenBioLLM-8B (Hybrid) | 92.02 | 0.63 | 0.97 | 0.98 | 0.63 | 0.76 |
| | LLama3.1-8B (Hybrid) | 93.7 | 0.77 | 0.98 | 0.91 | 0.77 | 0.83 |
| | LLama3.1-70B (Hybrid) | 92.52 | 0.75 | 0.97 | 0.87 | 0.75 | 0.8 |
| | LLama3.1-405B (Hybrid) | 92.2 | 0.84 | 0.94 | 0.79 | 0.84 | 0.82 |

Figure 6: Detail Metrics on Table 2

|  | Model | Accuracy | TPR | TNR | Precision | Recall | F1 |
|---|---|---|---|---|---|---|---|
| **Zero-shot** | OpenBioLLM-8B | 47.33 | 0.6 | 0.44 | 0.22 | 0.6 | 0.32 |
|  | LLama3.1-8B | 71.08 | 0.08 | 0.88 | 0.14 | 0.08 | 0.1 |
|  | LLama3.1-70B | 76.23 | 0.86 | 0.74 | 0.46 | 0.86 | 0.6 |
|  | LLama3.1-405B | 78.03 | 0.53 | 0.84 | 0.47 | 0.53 | 0.5 |
|  | MedAlpaca-7B | 23.1 | 0.04 | 0.28 | 0.01 | 0.04 | 0.02 |
|  | PMC-Llama-7B | 18.9 | 0.51 | 0.11 | 0.13 | 0.51 | 0.21 |
|  | Meditron-7B | 6.49 | 0.25 | 0.02 | 0.06 | 0.25 | 0.1 |
|  | Biomistral-7B | 31.79 | 0.52 | 0.27 | 0.15 | 0.52 | 0.24 |
|  | Phi3.5-MoE | 74.35 | 0.18 | 0.89 | 0.3 | 0.18 | 0.22 |
| **50-shots** | OpenBioLLM-8B | 57.71 | 0.57 | 0.58 | 0.26 | 0.57 | 0.36 |
|  | LLama3.1-8B | 80.19 | 0.32 | 0.93 | 0.53 | 0.32 | 0.4 |
|  | LLama3.1-70B | 81.8 | 0.37 | 0.93 | 0.59 | 0.37 | 0.46 |
|  | LLama3.1-405B | 85.12 | 0.46 | 0.95 | 0.72 | 0.46 | 0.56 |
|  | MedAlpaca-7B | 34.68 | 0.89 | 0.21 | 0.22 | 0.89 | 0.36 |
|  | PMC-Llama-7B | 19.7 | 0.35 | 0.16 | 0.1 | 0.35 | 0.15 |
|  | Biomistral-7B | 37.97 | 0.52 | 0.34 | 0.17 | 0.52 | 0.26 |
|  | Phi3.5-MoE | 79.6 | 0.7 | 0.82 | 0.5 | 0.7 | 0.58 |
| **Prompt-Assigned Zero-shot** | OpenBioLLM-8B (Hybrid) | 96.12 | 0.83 | 1.0 | 0.98 | 0.83 | 0.9 |
|  | LLama3.1-8B (Hybrid) | 94.31 | 0.8 | 0.98 | 0.92 | 0.8 | 0.85 |
|  | LLama3.1-70B (Hybrid) | 93.9 | 0.76 | 0.99 | 0.93 | 0.76 | 0.84 |
|  | LLama3.1-405B (Hybrid) | 95.17 | 0.84 | 0.98 | 0.92 | 0.84 | 0.88 |
| **Original** | OpenBioLLM-8B (Hybrid) | 96 | 0.83 | 0.89 | 0.94 | 0.83 | 0.91 |
|  | LLama3.1-8B (Hybrid) | 94 | 0.8 | 0.98 | 0.91 | 0.8 | 0.85 |
|  | LLama3.1-70B (Hybrid) | 94 | 0.76 | 0.99 | 0.95 | 0.76 | 0.85 |
|  | LLama3.1-405B (Hybrid) | 95 | 0.84 | 0.98 | 0.92 | 0.84 | 0.88 |
| **No Hierarchical Prompting** | OpenBioLLM-8B (Hybrid) | 86 | 0.66 | 0.91 | 0.65 | 0.66 | 0.65 |
|  | LLama3.1-8B (Hybrid) | 82 | 0.65 | 0.87 | 0.56 | 0.65 | 0.6 |
|  | LLama3.1-70B (Hybrid) | 89 | 0.66 | 0.95 | 0.77 | 0.66 | 0.71 |
|  | LLama3.1-405B (Hybrid) | 90 | 0.68 | 0.96 | 0.82 | 0.68 | 0.74 |
| **No CoT** | OpenBioLLM-8B (Hybrid) | 87 | 0.72 | 0.9 | 0.66 | 0.72 | 0.69 |
|  | LLama3.1-8B (Hybrid) | 79 | 0.71 | 0.81 | 0.49 | 0.71 | 0.58 |
|  | LLama3.1-70B (Hybrid) | 86 | 0.61 | 0.92 | 0.67 | 0.62 | 0.63 |
|  | LLama3.1-405B (Hybrid) | 83 | 0.72 | 0.86 | 0.58 | 0.72 | 0.64 |
| **No Hierarchical Prompting & No CoT** | OpenBioLLM-8B (Hybrid) | 83 | 0.63 | 0.88 | 0.58 | 0.63 | 0.6 |
|  | LLama3.1-8B (Hybrid) | 89 | 0.71 | 0.94 | 0.74 | 0.71 | 0.72 |
|  | LLama3.1-70B (Hybrid) | 87 | 0.59 | 0.94 | 0.72 | 0.59 | 0.65 |
|  | LLama3.1-405B (Hybrid) | 79 | 0.75 | 0.8 | 0.5 | 0.75 | 0.6 |

Figure 7: Abolition Study on 40 Factors

|  | Model | Accuracy | TPR | TNR | Precision | Recall | F1 |
|---|---|---|---|---|---|---|---|
| **Zero-shot** | OpenBioLLM-8B | 5.32 | 0.2 | 0.01 | 0.05 | 0.2 | 0.08 |
|  | LLama3.1-8B | 6.77 | 0.28 | 0.01 | 0.07 | 0.28 | 0.11 |
|  | LLama3.1-70B | 8.2 | 0.33 | 0.02 | 0.08 | 0.33 | 0.13 |
|  | LLama3.1-405B | 12.7 | 0.56 | 0.01 | 0.13 | 0.56 | 0.21 |
|  | MedAlpaca-7B | 4.07 | 0.08 | 0.03 | 0.02 | 0.08 | 0.03 |
|  | PMC-Llama-7B | 2.7 | 0.06 | 0.02 | 0.01 | 0.06 | 0.02 |
|  | Meditron-7B | 6.23 | 0.07 | 0.06 | 0.02 | 0.07 | 0.03 |
|  | Biomistral-7B | 5.1 | 0.18 | 0.02 | 0.05 | 0.18 | 0.07 |
|  | Phi3.5-MoE | 7.82 | 0.33 | 0.01 | 0.08 | 0.33 | 0.13 |
| **50-shots** | OpenBioLLM-8B | 37.2 | 0.28 | 0.4 | 0.11 | 0.28 | 0.15 |
|  | LLama3.1-8B | 37.4 | 0.87 | 0.25 | 0.23 | 0.87 | 0.36 |
|  | LLama3.1-70B | 44.93 | 0.48 | 0.44 | 0.18 | 0.48 | 0.27 |
|  | LLama3.1-405B | 53.74 | 0.28 | 0.6 | 0.15 | 0.28 | 0.2 |
|  | MedAlpaca-7B | 24.24 | 0.4 | 0.2 | 0.11 | 0.4 | 0.18 |
|  | PMC-Llama-7B | 13.3 | 0.37 | 0.07 | 0.09 | 0.37 | 0.15 |
|  | Biomistral-7B | 34.57 | 0.27 | 0.37 | 0.1 | 0.27 | 0.14 |
|  | Phi3.5-MoE | 31.7 | 0.8 | 0.19 | 0.2 | 0.8 | 0.32 |
| **Prompt-Assigned Zero-shot** | OpenBioLLM-8B (Hybrid) | 57.19 | 0.56 | 0.58 | 0.25 | 0.56 | 0.35 |
|  | LLama3.1-8B (Hybrid) | 55.23 | 0.25 | 0.63 | 0.15 | 0.25 | 0.19 |
|  | LLama3.1-70B (Hybrid) | 59.12 | 0.82 | 0.53 | 0.31 | 0.82 | 0.45 |
|  | LLama3.1-405B (Hybrid) | 79.0 | 0.49 | 0.87 | 0.49 | 0.51 | 0.49 |
| **Original** | OpenBioLLM-8B (Hybrid) | 0.26 | 0.28 | 0.58 | 0.26 | 0.56 | 0.35 |
|  | LLama3.1-8B (Hybrid) | 0.15 | 0.25 | 0.63 | 0.15 | 0.25 | 0.19 |
|  | LLama3.1-70B (Hybrid) | 0.31 | 0.82 | 0.53 | 0.31 | 0.82 | 0.45 |
|  | LLama3.1-405B (Hybrid) | 0.49 | 0.49 | 0.87 | 0.49 | 0.49 | 0.49 |
| **No Hierarchical Prompting** | OpenBioLLM-8B (Hybrid) | 0.24 | 0.49 | 0.59 | 0.24 | 0.49 | 0.32 |
|  | LLama3.1-8B (Hybrid) | 0.14 | 0.25 | 0.6 | 0.14 | 0.25 | 0.18 |
|  | LLama3.1-70B (Hybrid) | 0.28 | 0.7 | 0.54 | 0.28 | 0.7 | 0.4 |
|  | LLama3.1-405B (Hybrid) | 0.45 | 0.48 | 0.85 | 0.45 | 0.48 | 0.46 |
| **No CoT** | OpenBioLLM-8B (Hybrid) | 0.23 | 0.45 | 0.6 | 0.23 | 0.45 | 0.3 |
|  | LLama3.1-8B (Hybrid) | 0.13 | 0.24 | 0.6 | 0.13 | 0.24 | 0.17 |
|  | LLama3.1-70B (Hybrid) | 0.29 | 0.72 | 0.53 | 0.29 | 0.72 | 0.41 |
|  | LLama3.1-405B (Hybrid) | 0.44 | 0.48 | 0.84 | 0.44 | 0.48 | 0.46 |
| **No Hierarchical Prompting & No CoT** | OpenBioLLM-8B (Hybrid) | 0.21 | 0.43 | 0.58 | 0.21 | 0.43 | 0.28 |
|  | LLama3.1-8B (Hybrid) | 0.14 | 0.25 | 0.61 | 0.14 | 0.25 | 0.18 |
|  | LLama3.1-70B (Hybrid) | 0.28 | 0.7 | 0.54 | 0.28 | 0.7 | 0.4 |
|  | LLama3.1-405B (Hybrid) | 0.46 | 0.47 | 0.86 | 0.46 | 0.47 | 0.47 |

Figure 8: Abolition Study on 78 Factors

| Characteristics | All (n=7,199) | Non-PROM (n=5,716) | PROM (n=1,483) | P-Value |
|---|---|---|---|---|
| Age (years) | 29.16 (4.27) | 29.23 (4.29) | 28.88 (4.17) | 0.005 |
| Pre-pregnancy BMI (kg/m²) | 21.61 (2.97) | 21.63 (2.99) | 21.52 (2.90) | 0.22 |
| Mid-pregnancy BMI (kg/m²) | 24.42 (3.04) | 24.43 (3.06) | 24.38 (2.95) | 0.576 |
| Sleeping time (h) | 7.63 (1.07) | 7.64 (1.07) | 7.60 (1.07) | 0.162 |
| Education | | | | 0.523 |
|   Senior high school or below | 2818 (39.1) | 2255 (39.5) | 563 (38.0) | |
|   Junior college | 2538 (35.3) | 2011 (35.2) | 527 (35.5) | |
|   College degree or higher | 1843 (25.6) | 1450 (25.4) | 393 (26.5) | |
| Career | | | | 0.648 |
|   No | 2300 (31.9) | 1834 (32.1) | 466 (31.4) | |
|   Yes | 4899 (68.1) | 3882 (67.9) | 1017 (68.6) | |
| Area | | | | 0.09 |
|   Urban | 6531 (90.7) | 5203 (91.0) | 1328 (89.5) | |
|   Non-urban | 668 (9.3) | 513 (9.0) | 155 (10.5) | |
| Income (RMB/month) | | | | 0.36 |
|   ≤ 6000 | 6511 (90.4) | 5160 (90.3) | 1351 (91.1) | |
|   > 6000 | 688 (9.6) | 556 (9.7) | 132 (8.9) | |
| Outing method | | | | 0.854 |
|   Walking or cycling | 959 (13.3) | 766 (13.4) | 193 (13.0) | |
|   Electric vehicle | 1200 (16.7) | 957 (16.7) | 243 (16.4) | |
|   Private car or Public transportation | 5040 (70.0) | 3993 (69.9) | 1047 (70.6) | |
| Passive smoking | | | | 0.43 |
|   No | 4151 (57.7) | 3282 (57.4) | 869 (58.6) | |
|   Yes | 3048 (42.3) | 2434 (42.6) | 614 (41.4) | |
| Number of pregnancies | | | | <0.001 |
|   1 | 2640 (36.7) | 2006 (35.1) | 634 (42.8) | |
|   2 | 2466 (34.3) | 1983 (34.7) | 483 (32.6) | |
|   ≥ 3 | 2093 (29.1) | 1727 (30.2) | 366 (24.7) | |
| Time to Pregnancy (month) | | | | 0.481 |
|   ≤ 3 | 4777 (66.4) | 3781 (66.1) | 996 (67.2) | |
|   > 3 | 2422 (33.6) | 1935 (33.9) | 487 (32.8) | |
| Night shift | | | | 0.269 |
|   No | 6903 (95.9) | 5489 (96.0) | 1414 (95.3) | |
|   Yes | 296 (4.1) | 227 (4.0) | 69 (4.7) | |
| Physical activity (days/week) | | | | 0.695 |
|   < 3 | 4038 (56.1) | 3199 (56.0) | 839 (56.6) | |
|   ≥ 3 | 3161 (43.9) | 2517 (44.0) | 644 (43.4) | |
| Sedentary behavior (h/day) | | | | 0.788 |
|   < 2 | 911 (12.7) | 717 (12.5) | 194 (13.1) | |
|   2-6 | 3237 (45.0) | 2580 (45.1) | 657 (44.3) | |
|   ≥ 6 | 3051 (42.4) | 2419 (42.3) | 632 (42.6) | |
| Depression at mid-pregnancy | | | | 0.256 |
|   No | 4831 (67.1) | 3817 (66.8) | 1014 (68.4) | |
|   Yes | 2368 (32.9) | 1899 (33.2) | 469 (31.6) | |

Figure 9: Statistical description For the Cohort