# OpenReview forum: "No Factor Left Behind: Towards arbitrary amount of factors in the medical cohort analysis"
_ICLR.cc/2025/Conference — ICLR 2025 Conference Withdrawn Submission_

### Official Review · Reviewer_WVYT · 2024-10-22

**Soundness:** 2
**Presentation:** 2
**Contribution:** 3
**Rating:** 3
**Confidence:** 3

**Summary:**

This paper states an approach for improving medical cohort analysis using LLMs, especially by incorporating low-frequency factors that are usually not considered in traditional methods. They propose a method for developing prompts via both manual and automatic techniques, using methods such as MLM and cohort-specific prompts to improve risk prediction in medical data. The study explores the use of models such as LLama3.1 and MedAlpaca, among others, and evaluates their performance on different factor sets, showing better accuracy, especially in the prediction of PROM. Key contributions include the introduction of a factor-based interaction map to represent the relationships between individual factors and architectural improvements designed to reduce overfitting. The results show that the proposed method outperforms traditional supervised baselines, which provides a better tool for medical cohort analysis while addressing challenges in the LLM-based models.

**Strengths:**

Originality: This paper introduces an approach by focusing on low-frequency factors in medical cohort analysis, which are often ignored. Using manual and automatic prompt generation with LLMs like LLama3.1 and MedAlpaca is relatively creative, as is the factor-based interaction map.
Quality: The paper provides partial technical depth, with some explanations of the prompt generation techniques and factor-based interaction map.
Clarity: The paper is primarily clear, particularly in describing the prompt generation and the interaction map.
Significance: By incorporating low-frequency factors, the paper has the potential to significantly improve risk prediction in healthcare, such as in the prediction of premature rupture of membranes.
Overall comment on strengths: This paper makes a good contribution, especially in applying LLMs to low-frequency factors in healthcare.

**Weaknesses:**

The paper doesn’t compare the mentioned method with traditional machine learning models, which would highlight the specific advantages of the proposed approach. The computational cost and feasibility of automatic prompt generation aren’t discussed, leaving doubts about its practicality in clinical settings. The experiments are based on only one dataset, restricting the generalizability of the findings. Also, overfitting is a risk with the factor-based interaction map, but this paper lacks a discussion on a few mitigation strategies, such as regularization techniques. Some terms throughout the paper are not clear and need further clarification for non-expert readers. The explanations for evaluation metrics are too narrow and centered around accuracy without reporting other critical metrics such as AUC. Some ethical concerns and potential biases in the data are mentioned but not completely addressed and discussed.

**Questions:**

-The abstract section mentions 79% accuracy with 78 factors and 96% with 40 factors but doesn't give a context on how these results compare to benchmarks or their clinical significance. The term "numerical interpretable evidence" is unclear.
-In the introduction section, the paper states that traditional studies overlook most of the factors but doesn't elaborate on why adding so many factors is necessary or even how this can lead to better outcomes.
-The factor-based interaction map does not explain why it was chosen and how it improves performance. There is also no discussion on scalability, overfitting, or managing complexity as more factors are added.
-The automatic prompt generation method's scalability is mentioned, but computational costs and feasibility for clinical use are missing. There's no mention of time or resources required, making it hard to assess practical viability.
-The results rely on outdated models like LLama3.1 and MedAlpaca without clear justification. The reported accuracy metrics lack context and comparison to modern benchmarks, making it difficult to interpret their significance.
-There are no error bars or confidence intervals, making the reliability of the results hard to assess. These are essential to determine how robust the findings are in medical contexts where consistency is critical.
-The ethical concerns around using cohort-specific medical data are barely discussed despite the sensitivity of such data. More focus is needed on privacy risks and potential biases in using large language models on this data.
-The conclusion overstates the method's applicability, claiming it provides a universal solution despite showing only incremental improvements. The method's broader relevance to other conditions or datasets isn't convincingly demonstrated.
-Choosing 78 factors is not clearly mentioned. There's no explanation for why these factors were selected or how they were deemed necessary, making it so difficult to assess the significance of this approach.
-The manual prompt engineering process is impractical for large-scale settings or dynamic settings. Its lack of generalizability across conditions or datasets is a significant limitation, and the adaptability of these prompts in new domains isn't discussed.
-The automatic prompt generation pipeline lacks discussion on potential errors or biases, especially in medical applications where accuracy is crucial. The risks of generating irrelevant content aren't addressed.
-The interaction map may not scale well as more factors are added. As complexity increases, overfitting becomes a risk, but the paper doesn't discuss how to manage this or mitigate it such as regularization techniques. This subject cannot be ignored.
-Bias and fairness issues in the data aren't addressed. Without steps to ensure fairness, the model may propagate existing biases in the training data, leading to unequal outcomes for different patient groups.
-The cohort-specific prompts could be overfitted to the dataset and fail to generalize to new cohorts. There's no discussion on how adaptable these prompts are across different settings.
-The computational cost of fine-tuning large models isn't considered. Training these models is resource-heavy, and the paper doesn't quantify the time or hardware needed, limiting its practicality in clinical environments.
-The baseline comparison is only with LLMs, but a comparison to simpler machine learning models would be useful. Traditional models are often preferred in healthcare for their interpretability, and their inclusion would provide a clearer picture of the method's advantages.
-The dataset is from a specific cohort, but the findings' generalizability to other datasets is not discussed. Testing on diverse datasets would strengthen the paper's claims.

---

> ### Author Response · Authors · 2024-11-25
> **Reply to reviewer concerns**
>
> Thanks for your comment.
>
> Your questions and concerns help us to improve the whole study and optimize our study representations.
>
> Q1:  Our study doesn't give a context on how these results compare to benchmarks or their clinical significance
> A1: Yes, this question is critical to our method, and we want to test our method in the other benchmark and dataset.  However, to our knowledge, it is like the Medical Information Mart for Intensive Care[MIMIC(I-III)], UKBiobank, Global Burden of Disease(GBD), and the National Health and Nutrition Survey(NHANES).  We have very difficulty finding the same scale dataset that we collected for our cohort, which has more than 70 factors to one consequence.  The UKB has the most counted factors for the Body Mass Index(BMI), which has approximately 21 factors.  As for the MIMIC, the most counted factor is the electroencephalogram (EEG), which has approximately 35 factors.  The GBD, a classical and well-trusted database for epidemiology, is the most counted factor for total cancer
> rate, which has approximately 30 factors.  Last, the NHANES, since this is the cross-sectional study count in 2-3 years as a cycle(15 cycles, 30 years), has the most counted factor for lung function test, which has 45 factors.  We are sorry for not testing those datasets since we have limited computing resources and are far from the counts of factors we collected.
>
> Q2: The factor-based interaction map does not explain why it was chosen and how it improves performance.  here is also no discussion on scalability, overfitting, or managing complexity as more factors are added.
> Q2: Thanks for your comment.  This process is based on game theory to annotate pure numerical data to help LLM understand whether this feature and its value have a positive or negative effect on the consequence, case by case.  We are sorry not to discuss the scalability and complexity since we can't find larger counts of factors before we submit this manuscript.  As for the overfitting and managing complexity, we are sorry that we are not representing the training process of this tree-based generator.  We will add this to the appendix.
>
> Q3: -The automatic prompt generation method's scalability is mentioned, but computational costs and feasibility for clinical use are missing.  There's no mention of time or resources required, making it hard to assess practical viability.
> A3: We are very grateful that the reviewer is concerned about the insufficient computation resources in the current BioMed area.  We can't represent another medical area since we focus on maternal and infant health(MIH).  To our knowledge, fine-tuning a BERT-like model to be adapted in MIH has worked sufficiently.  Due to the privacy restrictions in the cohort regulation, it is difficult to transport any non-public cohort data to data centers and cloud platforms to train the Flan-T5 scale model or PEFT the open source model.  We fine-tuned the PubMedBERT in 11 days with RTX 4090.  We will open-source both the fine-tuned code and the MIH-related paper dataset shortly.
>
> Q4: The results rely on outdated models like LLama3.1 and MedAlpaca without clear justification.  The reported accuracy metrics lack context and comparison to modern benchmarks, making it difficult to interpret their significance.
> A4: Thanks for your comments.  We are sorry to have selected the outdated models since we can only use the local computational resources.  We also have noticed that ANTHROPIC released a statistical approach to model evaluations.  We are working on this approach to providing modern benchmarks to AI4Cohort area.  Sorry again, we are make reviewer to feel difficult to interpret significance.
>
> Q5: There are no error bars or confidence intervals, making the reliability of the results hard to assess.  These are essential to determine how robust the findings are in medical contexts where consistency is critical.
> A5: Thanks for your comment.  As answer 4 mentioned, we are working on the statistical approach to model evaluations provided by ANTHROPIC.  Before we noticed this approach, we were sorry that we think too simple to evaluate those models, though we had searched the related research area.
>
> Q6: The ethical concerns around using cohort-specific medical data are barely discussed despite the sensitivity of such data.  More focus is needed on privacy risks and potential biases in using large language models on this data.
> A6: Thanks for your comment.  Cohort data sensitivity and participant privacy are also among the issues of concern.  All cohort data is restricted to local storage, and training data consists of unidentified-unique ID with feature value.  The whole training process is not related to any personal information that could identify the person.  We are also sorry that we can't represent the Ethical Approment since ICLR is not allowed to present any information that could identify authors.

---

> > ### Author Response · Authors · 2024-11-25
> > **Reply to reviewer concerns Part2**
> >
> > Q7: Choosing 78 factors is not clearly mentioned. There's no explanation for why these factors were selected or how they were deemed necessary, making it difficult to assess this approach's significance.
> > A7: Thanks for your comment. As we answer in the Q1. For the traditional risk assessment approach, the researchers are only focused on the factors mentioned in the golden sample questionnaires. Therefore, other factors will be ignored, and the risk assessment frequently fails in the PROM risk evaluation. Based on this motivation, we are trying to collect as many factors as possible to build the larger landscape for PROM with the LLM to discover the risk factors case by case.
> >
> > Q8:The manual prompt engineering process is impractical for large-scale settings or dynamic settings. Its lack of generalizability across conditions or datasets is a significant limitation, and the adaptability of these prompts in new domains isn't discussed.
> > A8: Thanks for your comment. We are sorry that we are not mentioned. We only write 100 prompts in the manual. We are grateful that the reviewer was concerned about the limitations. We will try to apply this approach in oncology using mRNA data to let LLMs predict drug response.
> >
> > Q9: The automatic prompt generation pipeline lacks discussion on potential errors or biases, especially in medical applications where accuracy is crucial. The risks of generating irrelevant content aren't addressed.
> > A9: Thanks for your comment. We have deployed the guard LLM model( llama guard-7b) during the automatic generation, and since this is a retrospective cohort experiment, we are not applying human expert checking case by case. We are applying human expert checking in the following real-world testing, but it is not included in the article. Thanks for your reminder. We will address the hallucinations and misinformation in the next version of our article.
> >
> > Q10: The interaction map may not scale well as more factors are added. As complexity increases, overfitting becomes a risk, but the paper doesn't discuss how to manage this or mitigate it such as regularization techniques.
> > A10: Thanks for your comment. As we mentioned in answer 1, there may not be larger counts of factors as we performed in the current PubMed database(We have double-checked it). As complexity increases, we will consider using the graph to represent the participant as this method worked in the recommended system, and we will try to collect more factors than 78 to test this hypothesis.
> >
> > Q11:This subject cannot be ignored. -Bias and fairness issues in the data aren't addressed. Without steps to ensure fairness, the model may propagate existing biases in the training data, leading to unequal outcomes for different patient groups.
> > A11: Thanks for your comment. The basis and fairness concern is one of the issues we have been most concerned about since we researched MIH for decades. We have been considering for a long time how to represent fairness and impartiality without going against the double-blind review procedure, in which we can't display either detailed ethical approval or detailed generations for unidentified participants to prove that the model has been ensured fairness. We are thankful for your concern, which will be solved as we submit the detailed ethical approval for the next version of our article.
> >
> > Q12: The cohort-specific prompts could be overfitted to the dataset and fail to generalize to new cohorts. There's no discussion on how adaptable these prompts are across different settings.
> > A12: Thanks for your comment. Since our training process aims to update the current MIH knowledge, we are not training the detailed cohort feature value. There is a slight chance of overfitting on the related paper. All the LLMs inference settings are defaulted in the Pytorch framework. We gratefully thank the reviewer for reminding our representation may mislead the reader that we are training the LLMs on cohort data.
> >
> > Q13: The computational cost of fine-tuning large models isn't considered. Training these models is resource-heavy, and the paper doesn't quantify the time or hardware needed, limiting its practicality in clinical environments.
> > A13: Thanks for your comment. We are very grateful that the reviewer is concerned about the insufficient computation resources in the current BioMed area. We are using RTX 4090 Dual for PEFT the OpenBiollama in 15 days.
> > We are really understanding the  insufficient computation resources in clinical environments, but we believe two RTX 4090  has practicality in the most of hospitals
> > We are sorry about the missing representations. We will add those details to the appendix in the next version of our article.

---

> > > ### Author Response · Authors · 2024-11-25
> > > **Reply to reviewer concerns Part3**
> > >
> > > Q14: The baseline comparison is only with LLMs, but a comparison to simpler machine learning models would be useful. Traditional models are often preferred in healthcare for their interpretability, and their inclusion would provide a clearer picture of the method's advantages.
> > > A14: Thanks for your comment. We compare LLM performance with Random forest, XGBoost, LightGBM, and Logistic regression. We are also choosing the support vector machine, but it hasn't finished training yet, as the deadline is approaching.
> > > The dataset is split into an 80% training set and a 20% test dataset; before the split, it was balanced by the SMOTH method.
> > > The XGBoost was the most accurate, surpassing the LLM approach with a 97.9% accuracy in 40 factors in the test dataset.
> > > The Random Forest has the best accuracy among the four models in 78 factors and has 73.2%  accuracy in the test dataset.
> > > The most traditional method, Logistic regression, is 52% accurate in 40 factors and 51.4% accurate in 78 factors.
> > > We are grateful to the reviewer for reminding us to add the comparison to other machine-learning models.
> > > We will add those results in detail to the appendix in the next version of our paper.
> > >
> > > Q15: The dataset is from a specific cohort, but the findings' generalizability to other datasets is not discussed. Testing on diverse datasets would strengthen the paper's claims.
> > > A15: Thanks for your comments. We are willing to test this method in the dataset, as we mentioned in answer 1. However, our initial aim is to enlarge the factor counts in the cohort analysis that were never reached before inference by LLMs. We are sorry we don't have sufficient time to collect the related paper to become training tokens to update the LLMs' knowledge like what we performed in the MIH for the UKBiobank, NHANES and, GBD, etc. We will continue to acclimate more up-to-date medical knowledge in different areas to test the generalizability. Lastly, we would be grateful if you would consider the answer above to evaluate our research. Your questions will be addressed in the next version of our article.

---

### Official Review · Reviewer_rtzB · 2024-11-01

**Soundness:** 2
**Presentation:** 1
**Contribution:** 2
**Rating:** 3
**Confidence:** 2

**Summary:**

The text does not flow well and I feel like sometimes sentences do not logically follow each other. The text is poorly formatted and a lot of details are missing. I have to say that the language is formatted in such a way that I am probably missing the the point of the paper. It feels like large parts have been written/corrected by an LLM. This is the case in the entire text.

**Strengths:**

- Used existing biomedical models in experiments.
- Collected a new dataset to use in the experiments (details are missing about this dataset, though).

**Weaknesses:**

- line 275: why is this seperate line here?
- 402: spelling error: superviesd
- Figure 2: the text overlaps with the bars. Also predication is meant to be prediction?
- line 486: is supposed to be a section heading?
- Line 522: multiple links/garbled (?) text
- No contextualization/related work of PROM which is only fully mentioned in the abstract. This should be part of the related work section and then again in the discussion section.
- Line 151: why cite the SHAP paper?
- The PROM use case is not explained well, and it is not clear how this would contribute to the wider ICLR community. There needs to be a generalization of the knowledge.

**Questions:**

- I don't understand the "husband education level" example for a biomedical LLM. Why would this be relevant to the task(s) that you are trying to solve? Again, the lack of related work and the explanation of the application could help here.

---

> ### Author Response · Authors · 2024-11-25
> **Reply to reviewer concerns**
>
> Thanks for your comment
>
> We are sorry that latex's green hand usage challenged your interpretation of our study.  We respect your efforts to review our work.
>
> Q1: Line 151: why cite the SHAP paper?
> A1: Thanks for your comment.  The factor-based interaction map is developed based on the part of the SHAP interpretable function, which also uses game theory to understand the model and data.  Therefore, to maintain basic respect, we cite the SHAP paper naturally.
>
> Q2: I don't understand the "husband education level" example for a biomedical LLM.  Why would this be relevant to the task(s) that you are trying to solve?  Again, the lack of related work and the explanation of the application could help here.
> A2: Thanks for your comment. Husband's education level is a traditional risk factor for pregnant women. We want to utilize this feature annotation process to represent how Generation based on Medical Language Model worked.
> We are sorry about the lack of further explanation.
> Traditional PROM risk assessments rely solely on predefined questionnaire factors, overlooking potential contributors and leading to frequent inaccuracies.  To address this limitation, we are utilizing a Large Language Model (LLM) to comprehensively identify a broader range of risk factors on a case-by-case basis, creating a more complete understanding of PROM risk.
> We appreciate the reviewers' insightful comments and valuable suggestions regarding the validation of our method on benchmark datasets.  We acknowledge the concern regarding using our dataset rather than widely adopted benchmarks like MIMIC, UKBiobank, GBD, or NHANES.
> The primary challenge in utilizing these public datasets stems from the significantly higher dimensionality of non-public cohorts like we collected.  Our data features over 70 factors contributing to a single consequence.  In contrast, the publicly available datasets we examined exhibit considerably lower factor counts for their respective consequences: UKBiobank (maximum of 21 factors for BMI), MIMIC (maximum of 35 factors for EEG), GBD (maximum of 30 factors for total cancer rate), and NHANES (maximum of 45 factors for lung function tests).  This substantial difference in dimensionality poses a significant obstacle to directly applying our method to these benchmark datasets.  Our approach is specifically designed to handle the complexity inherent in high-dimensional data, and scaling the factor count more than the benchmark datasets would potentially compromise the integrity and interpretability of risk access.
> Furthermore, our limited computing resources currently prevent us from processing the massive scale of datasets like MIMIC or UKBiobank in their entirety.  This constraint restricts our ability to test our method on these resources now.  We acknowledge that these limitations may impact the generalizability of our findings.  However, we believe that the unique characteristics of our dataset, specifically the high factor count, present a practical method to explore the relationships between a large number of variables and the outcome of interest in the other same-scale non-public cohorts.
>
> Q3: It feels like large parts have been written/corrected by an LLM
> A3: Thanks for your comment.  We are sorry we are missing a declaration of Grammarly usage for Polish purposes.  This declaration will be added in the next version of the article.
>
> We respectfully submit the above response for your evaluation of our research.
> A revised version of the article will incorporate your questions.
>
> Again, I'm sorry for our green hand latex usage.

---

> > ### Comment · Reviewer_rtzB · 2024-11-26
> > **Reply to rebuttal**
> >
> > Dear Authors,
> >
> > Thank you for your (late) reply to my review. I would suggest, for your next submission, getting feedback from a Native English speaker to make sure what you have written accurately represents what you did. Moreover, since I have not seen an improved version of the paper, I will keep my score. Thanks to everyone for their efforts.

---

### Official Review · Reviewer_YLQt · 2024-11-07

**Soundness:** 2
**Presentation:** 1
**Contribution:** 2
**Rating:** 3
**Confidence:** 3

**Summary:**

This study aims to enhance predictive modeling for medical risk assessment using large language models (LLMs) by addressing the challenge of integrating a broad range of low- and high-frequency factors. Through expert interviews and specialized data mining, the team developed a model capable of accurate PROM risk assessment across 70 factors, achieving 79% accuracy on 78 factors and 96% accuracy on 40 factors.

**Strengths:**

+ The study overall plan (while fairly unclear) seems fairly reasonable and timely.
+ The study aims to use an LLM-based pipeline for a novel application (i.e., identifying the factors to include in medical cohorts), in a customized and expanded way.
+ The study adopts a few solid theoretical frameworks, such as the one from epidemiology.
+ The study includes a user (human) study, although its design and findings are a bit unclear.

**Weaknesses:**

+ Despite the excitement, the primary concern about this submission is about its presentation and soundness. In the submitted format, the overall rationale and design seems quite convoluted and unclear. The whole paper's flow is weak. Numerous formatting and typos can be seen in the document. For instance, the whole paragraph related to Equation 4 seems to be duplicated. Or the text refers to the left and right side of Fig 1, but it seems that the authors were referring to the top and bottom parts. It is very hard to understand Section 3.2 (a key section for the Method) and its connection to 3.3 is unclear. Unfortunately, this issue makes evaluation of the core concepts and contributions very challenging.

**Questions:**

+ How function v in Eq 4 (or 5) is implemented in practice? That seems to have a key role in the process.
+ How the expert-augmented part is implemented? Are they sourced from the human participants? What would this mean for a general application of this tool?

---

> ### Author Response · Authors · 2024-11-25
> **Reply to reviewer concerns**
>
> Thanks for your comment
>
> We are sorry that latex's green hand usage made you feel challenged to interpret our study.
> We respect your efforts on the question about our work.
>
> W1: Numerous formatting and typos can be seen in the document.
> A1: Thanks for your patience. We will try to fix all of the formatting and typos in the next version.
>
> Q1:  How function v in Eq 4 (or 5) is implemented in practice? That seems to have a key role in the process.
> A1: Thanks for your comment. We are sorry about misrepresentation of equations. This process is implemented in Python to let OpenBioLLama (OBL) annate one feature and its value while the Eq5 is the loop that OBL annate all of the features that a participant owns.
>
> Q2: How is the expert-augmented part implemented? Are the experts sourced from the human participants? What would this mean for the general application of this tool?
> A2: Thanks for your comment. The expert-augmented part is implemented by training OBL on the up-to-date tokens from the domain-specific papers. Yes, the experts are sourced from human participants; since the papers are written by human experts and peer-reviewed, those papers also become the reference for human experts in clinic decision-making.
>
> We are thanks again for your patience due to our formatting and typos.
>
> Lastly, we would be grateful if you would consider the answer above to evaluate our research. Your questions will be addressed in the next version of our article.

---

### Note · Authors · 2024-11-27

**Comment:**

After the rebuttal, all authors agree to withdraw the current version of NFLB.

We are grateful for the reviewers' efforts to help us improve our study and representation. However, due to our lack of experience at the conference, we apologize for the late reply. We especially thank the Reviewer WVYT, for providing most of the constructive suggestions.

We will refine NFLB in the following steps:
1. Taking experiments in the MIMIC, UKB, GBD, and NHANES with NLFB
2. Optimizing the language representation and overall flow of our manuscript
3. Taking Modern Benchmarks to evaluate the NFLB comprehensively.

The next version of NFLB will be updated soon.

Again, many thanks to the Reviewer WVYT.

**Withdrawal Confirmation:**

I have read and agree with the venue's withdrawal policy on behalf of myself and my co-authors.